



# Occurrence of new particle formation events in Siberian and Finnish boreal forest

Helmi Uusitalo[1], Jenni Kontkanen[1], Ilona Ylivinkka[1,2], Ekaterina Ezhova[1], Anastasiia Demakova[1], Mikhail Arshinov[3], Boris D. Belan[3], Denis K. Davydov[3], Nan Ma[4], Tuukka Petäjä[1], Alfred Wiedensohler[5], Markku Kulmala[1,6,7], and Tuomo Nieminen[1,8]

[1]Institute for Atmospheric and Earth System Research (INAR) / Physics, Faculty of Science, University of Helsinki, Finland
[2]Station for Measuring Ecosystem - Atmosphere Relations II (SMEAR II), University of Helsinki, Korkeakoski, Finland
[3]V. E. Zuev Institute of Atmospheric Optics, SB RAS, Tomsk, Russia
[4]Institute for Environmental and Climate Research, Jinan University, Guangzhou, China
[5]Leibniz Institute for Tropospheric Research, Leipzig, Germany
[6]Aerosol and Haze Laboratory, Beijing Advanced Innovation Center for Soft Matter Science and Engineering, Beijing University of Chemical Technology, Beijing, China
[7]Joint International Research Laboratory of Atmospheric and Earth System Sciences, Nanjing University, Nanjing, China
[8]Institute for Atmospheric and Earth System Research (INAR) / Forest Sciences, Faculty of Agriculture and Forestry, University of Helsinki, Finland

**Correspondence:** Tuomo Nieminen (tuomo.nieminen@helsinki.fi)

**Abstract.** The occurence of new particle formation (NPF) events was investigated at four sites in the boreal forest environment (Hyytiälä SMEAR II and Värriö SMEAR I in Finland; Tomsk-Fonovaya and ZOTTO in Siberia, Russia), by analyzing measured particle number size-distributions (PNSD) and theoretical calculations of particle survival probabilities. NPF events were less frequent at the Siberian sites than at the Finnish sites. This is likely linked to lower survival probabilities of the

freshly-formed particles at the Siberian sites, due to higher coagulational losses and lower particle growth rates. Another factor affecting the frequency of observed NPF events is the minimum detectable particle size. When the NPF event classification was made for Hyytiälä, Värriö and Tomsk-Fonovaya sites based on PNSD starting from 15 nm instead of 3 nm, the observed NPF frequencies decreased. This result highlights the importance of measuring PNSD starting from sub-10 nm particles, in order to obtain reliable estimates of the NPF characteristics.

# 1   Introduction

Atmospheric aerosol particles play a significant role in the climate system and they also diminish the air quality and are harmful to human health (Pöschl, 2005; Nel, 2005; Lelieveld et al., 2015). Aerosol particles affect the climate in two ways. Firstly, they can directly scatter and absorb solar radiation. When aerosol particles scatter radiation they tend to cool the climate, whereas absorbing particles, such as soot, are warming the climate. Secondly, aerosol particles affect the climate also indirectly by acting

as cloud condensation nuclei (CCN) and thus changing the reflectance and lifetime of clouds (Boucher et al., 2013; Rosenfeld et al., 2014; Haywood, 2016). According to the Intergovernmental Panel on Climate Change (IPCC), aerosol particles and clouds are the largest uncertainty factor when studying climate change (Boucher et al., 2013), but the knowledge of particle



formation and behavior is still relatively low. Therefore, increased understanding on atmospheric aerosol particles and their interactions with clouds is essential when concentrating on mitigating climate change.

The sizes of aerosol particles vary from a few nanometers to hundreds of micrometers. Earlier, commonly used aerosol measurement instruments were able to observe only particles larger than 3 nm but with the instruments developed in recent years, even smaller than 1 nm particles can now be measured (Vanhanen et al., 2011; Kangasluoma et al., 2020). In order to affect the climate, the particles have to grow to the size of about 50-100 nm. Nowadays, the studies of below 3 nm particles have become more common, but there are still large measurement uncertainties and differences between measurement instruments

(Kontkanen et al., 2017; Kangasluoma et al., 2020).

New particle formation (NPF) is a globally significant phenomenon contributing to the number concentration of aerosol particles. In NPF, molecular clusters are formed by nucleation from atmospheric vapors and after that they grow to larger sizes mainly by condensation of vapours (Kulmala et al., 2013, 2014). NPF events (i.e. formation and the subsequent growth of new 3-25 nm particles) have been observed in various environments all around the world (Kerminen et al., 2018; Nieminen et al.,

2018). Many environmental conditions have been observed to have an influence on atmospheric NPF, including meteorological conditions and the concentrations of nucleation precursor vapors (Dada et al., 2017). One of the most significant factors is the concentration of sulphuric acid, which has been observed to be linked to the formation of the smallest 1-2 nm clusters and the frequency of NPF events (Kulmala et al., 2013; Nieminen et al., 2014). In addition to sulphuric acid, the clusters need stabilizing base molecules, such as amines and ammonia (Almeida et al., 2013), and organic molecules (Schobesberger et al.,

2013). Oxidation products of various volatile organic compounds have a significant role in the particle growth to larger sizes (Donahue et al., 2013; Ehn et al., 2014; Mohr et al., 2019). During the particle growth, the freshly formed particles are also subject to coagulational scavenging by the larger pre-existing aerosol. Therefore, low coagulation sink is favourable for a larger fraction of the nucleated particle reaching CCN sizes (Kulmala et al., 2017).

In boreal forest NPF events are most frequent during spring time (Nieminen et al., 2018). In the SMEAR II station (Station for

Measuring Ecosystem-Atmosphere Relations) in Hyytiälä in Finland, there are typically 60-120 event days per year, of which the particle growth rate can be reliably determined for about 30-40 events per year (Nieminen et al., 2014). Somewhat smaller NPF frequencies of around 15% of days annually are observed in Värriö SMEAR I station (Kyrö et al., 2014). Wiedensohler et al. (2019) analyzed in their study particle number size distribution (PNSD) data measured in Zotino Tall Tower Observatory (ZOTTO) in Siberia area. They observed clear NPF events on only 11 days during three-year measurement period, which is

significantly less than at the SMEAR II station. The low occurrence of nucleation mode particles in ZOTTO had been reported also in earlier study by (Heintzenberg et al., 2011). There is no clear reason for infrequent new particle formation in ZOTTO station so far.

In this work, new particle formation in Siberian and Finnish boreal forest environment is studied utilizing PNSD measurement results and theoretical calculations. We use PNSD data from four measurement stations in the boreal forest region:

Hyytiälä SMEAR II and Värriö SMEAR I stations in Finland, Fonovaya station in Tomsk region (West Siberia), and Zotino Tall Tower Observatory in East Siberia, Russia. Our objective is to find out why new particle formation is infrequently observed in Zotino Observatory compared to NPF observations at other measurement stations in the boreal area (especially in Hyytiälä





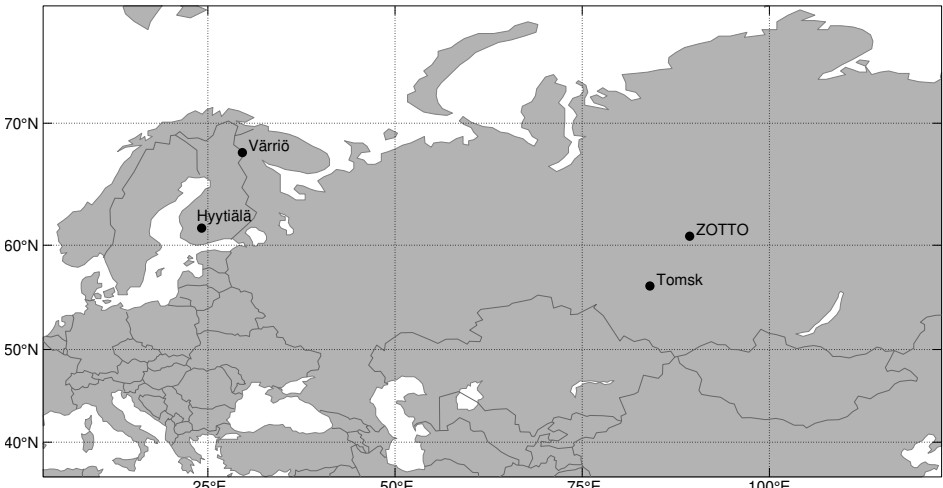

**Figure 1.** The locations of the measurement sites.

and Värriö). We also investigate how the minimum particle size in PNSD measurements affects the NPF analysis results. We use measured PNSDs to calculate particle growth rates, formation rates and condensation sinks to compare NPF event charac-

teristics in these four measurement stations. We also utilize theoretical equations describing particle formation rate of different particle sizes and particle survival probability.

## 2 Methods

### 2.1 Measurement sites

In this work, we use the measurements from four measurement sites located in the boreal forest zone: Hyytiälä SMEAR II and

Värriö SMEAR I stations in Finland, and Fonovaya measurement station near Tomsk and and Zotino Tall Tower Observatory in Russia (Fig. 1).

Hyytiälä measurement station (61°51' N, 24°17' E) is located 181 m above sea level and 190 km north-west of Helsinki. The nearest large city is Tampere 50 km to south-west from the station. The station is in rural environment and is surrounded by a 60 year old Scots pine dominated forest. At the station, comprehensive measurements of meteorological conditions, trace gas

concentrations and aerosol particle number size distributions have been continuously performed since 1996 (Hari and Kulmala, 2005).

Värriö measurement station (67°45' N, 29°36' E) is located in Värriö strict nature reserve area in Salla, Finnish Lapland. It is 395 m above sea level on the top of the hill Kotovaara. The station is surrounded by a Scots pine dominated forest and can be considered a remote site far from anthropogenic pollution sources, except for occasional long-range transport from the

industrial areas in Kola peninsula in direction east to north-east (Kyrö et al., 2014). Continuous aerosol size-distribution, trace gas and meteorological measurements have been started at the station in 1997.

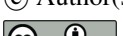



The Tomsk Fonovaya measurement station (56°25 N, 84°04 E, 80 m above sea level) is located in Siberia, 60 km west from the city of Tomsk on the shore of Ob river. The station is in rural area surrounded by a mixed forest of Scots pine, birch and aspen. The station has several instruments to measure aerosols, trace gases and meteorological conditions (Matvienko et al.,
75   2015).

The Zotino Tall Tower Observatory (ZOTTO; 60°80 N, 89°35 E, 180 m above sea level) is located in central Siberia about 500 km north-east from Tomsk. The forest surrounding the site is mostly Scots pines, which are 100 year old. In ZOTTO, the PNSDs are measured from the tall tower at the heights of 50 m and 300 m above ground level (Heintzenberg et al., 2011).

From Hyytiälä, Värriö and Tomsk-Fonovaya stations, we use PNSD data, and temperature ($T$) and relative humidity (RH)
data. From ZOTTO, we use the PNSD data. The data from Hyytiälä, Värriö and Tomsk-Fonovaya stations are from January 2017 to June 2019. The data from ZOTTO is from September 2006 to April 2009.

### 2.2   Particle size distribution measurements

In Hyytiälä and Värriö, the number size distribution of 3-1000 nm particles is measured with a twin-DMPS (Differential Mobility Particle Sizer) system (Aalto et al., 2001). The DMPS system consists of two differential mobility analyzers (DMA),
aerosol neutralizer and a condensation particle counter (CPC). With the neutralizer, aerosol particles can be charged and neutralized to form a known charge distribution. The DMA classifies particles to certain size ranges based on their electrical mobility. The CPC grows particles with condensation to the sizes where they can be detected and counted optically.

In Tomsk-Fonovaya station, the particle size distributions from 3 nm to 200 nm are measured with a diffusional particle sizer (DPS; Julanov et al., 2002). The DPS consists of the Novosibirsk-type diffusion battery and a condensation particle counter
(Grimm Model 5.403). Larger than 200 nm particle size distributions (data from 0.3 $\mu$m to 1 $\mu$m was used) are measured with optical particle counter (OPC, Grimm Model 1.108; Antonovich et al., 2018). The DPS data was averaged to the same time resolution with OPC data, in order to obtain the combined particle size-distribution in 3-1000 nm diameter range.

In ZOTTO, the PNSD data is measured with TROPOS-type mobility particle size spectrometer in the size range 10-835 nm. The mobility particle size spectrometer is described in more detail by Wiedensohler et al. (2012). The sample air is taken at the
heights of 50 m and 300 m above ground, and the resulting particle losses in the sample lines have been corrected according to Birmili et al. (2007).

### 2.3   New particle formation event classification

We classified the measurement period days at each site to three different classes: NPF event days, non-event days and undefined days. The classification was done by visually inspecting the time-evolution of particle size distribution on each day and
following the criteria by Dal Maso et al. (2005). If a new growing particle mode was observed in the nucleation mode (3-25 nm), the day was classified as an event day. If no new nucleation mode particles were observed, the day was classified as a non-event day. If nucleation mode particles were observed but they did not grow, or if particle growth was observed in Aitken mode (25-100 nm), the day was classified as an undefined day. We performed the event classification for each site by utilizing the whole measured particle size distribution, i.e. by studying larger than 3 nm particles in Hyytiälä, Värriö and Tomsk, and





larger than 15 nm particles in ZOTTO. In addition, we wanted to see if the measured size range can affect the result of the
classification, as earlier studies have shown that sometimes the NPF events can be observed clearly only at the smallest particle
sizes (Buenrostro Mazon et al., 2009; Dada et al., 2018). Therefore, we also performed the event classification for Hyytiälä,
Värriö and Tomsk sites by only studying particle sizes larger than 10 and 15 nm. For ZOTTO, the event classification was done
using PNSD data measured at the height of 50 m above ground. In addition, we compared the PNSD data measured at the

heights of 50 m and 300 m to see the possible differences in the occurrence of NPF events between these heights.

### 2.4 Determining particle growth rate and formation rate

Particle growth rate (GR) tells how fast the particle diameter increases with time during a new particle formation event. GRs
were determined by using a mode fitting method, where 1–3 log-normal modes were fitted to PNSD data at each time step.
The fitting was done using the automated algorithm by Hussein et al. (2005). Based on these fits, the geometric mean diameter

of the mode formed in a new particle formation event was determined for each time step (see Sect. 3.1 for examples). Finally,
GR was obtained as the slope of the linear fit to mode mean diameter vs time data.

Particle formation rate tells how fast the certain sized particles are formed. The unit of formation rate is typically given as
particles per cubic centimeter and second ($cm^{-3}s^{-1}$) and the studied particle size is shown in the subscript (e.g. $J_3$ for 3 nm
particles). The time-evolution of particle concentration in a size range with $[d_p, d_p + \Delta d_p]$ can be described with a balance

equation (Kulmala et al., 2012):

$$\frac{dN_{d_p}}{dt} = J_{d_p} - \text{losses},\qquad(1)$$

where $dN_{d_p}/dt$ is the observed change in the particle concentration in the studied size range, $J_{d_p}$ is the particle formation rate
and losses describe the loss rate of particles in that size range due to different processes (coagulation into pre-existing larger
particles and condensational growth out of the size range). Thus, particle formation rate can be obtained from

$$J_{d_p} = \frac{dN_{d_p}}{dt} + \text{CoagS}_{d_p} \cdot N_{d_p} + \frac{\text{GR}}{\Delta d_p} \cdot N_{d_p},\qquad(2)$$

where $N_{d_p}$ is the concentration of the particles in the studied size bin $d_p$, CoagS is the coagulation sink and GR is the growth
rate. Thus, the second term on the right-hand side describes the losses of particles in the studied size range due to coagulation
scavenging and the third term the losses due to growth out of the size range.

The coagulation sink can be determined from the measured particle size distribution for particle size $d_p$ from (Kulmala et

al., 2012):

$$\text{CoagS}_{d_p} = \sum_{d_p'=d_p}^{d_{p,\max}} K(d_p, d_p') N_{d_p'},\qquad(3)$$

where $K(d_p, d_p')$ is the coagulation coefficient between particles with diameters $d_p$ and $d_p'$. We calculated coagulation sinks us-
ing corrections for relative humidity dependent hygroscopic growth and temperature (based on the parameterization of Laakso
et al. (2004)), except in ZOTTO where only constant temperature (10 °C) was used due to lack of temperature measurements.





We determined particle formation rates from Eq. 2 and calculated their average value during new particle formation event times. Event time was determined to be the time period when particle concentration in the smallest measured size bin was elevated. Thus, when the minimum particle size is larger (for example 10 nm or 15 nm), the event time is possibly later than with smaller minimum particle size.

Formation rates were determined starting from the smallest detectable particle size, which is 3 nm for Hyytiälä, Värriö and
Tomsk. The minimum measured particle size in ZOTTO was 10 nm, but due to small particle concentrations between 10 nm and 15 nm, even during many of the NPF events, we assumed here that the minimum particle size was 15 nm.

### 2.5  Particle loss processes and survival probability

Condensation sink (CS) is the loss rate of vapors on pre-existing particle population, mainly determined by the surface area of the particles. CS can be calculated from measured particle size distribution according to (Kulmala et al., 2012):

$$\text{CS} = 2\pi D_v \sum_{d'_\text{p}} \beta_{m,d'_\text{p}} d'_\text{p} N_{d'_\text{p}} \qquad (4)$$

where $D_v$ is the vapour diffusion coefficient (assumed to be that of sulphuric acid) and $\beta_m$ is the transitional correction factor for the mass flux. For CS calculations, the PNSD data from 3 nm to 1 $\mu$m was used, except in ZOTTO where the whole available PNSD data from 10 nm to 840 nm was used.

Particle formation rates are typically calculated for the smallest measured particle size. The connection between the particle
formation rates at different sizes can be described with the equation (Kerminen and Kulmala, 2002)

$$J_2 = J_1 \cdot \exp\left(-0.23 \cdot \frac{\text{CS}'}{\text{GR}'}\left(\frac{1}{d_1} - \frac{1}{d_2}\right)\right), \qquad (5)$$

where $J_2$ is the formation rate of larger particle size, $J_1$ is the formation rate of smaller size particle and $d_1$ and $d_2$ are the diameters of the particles, respectively. In Eq. 5, the parameter CS' = CS (s$^{-1}$) / $4\pi \cdot 10^{-5}$ and GR' is GR (nm h$^{-1}$) or GR (m s$^{-1}$) / 3600 s $\cdot$ 10$^{-9}$ m.

The exponent term in Eq. (5) describes the probability of a particle to survive from the smaller size to the larger size, which depends on the competition between the particle growth and their loss due to scavenging by pre-existing particle surfaces (described by CS) (Kulmala et al., 2017). Let us define the survival probability of a particle between sizes $d_1$ and $d_2$ as the ratio of the formation rates:

$$\frac{J_2}{J_1} = \exp\left(-0.23 \cdot \frac{\text{CS}'}{\text{GR}'}\left(\frac{1}{d_1} - \frac{1}{d_2}\right)\right). \qquad (6)$$

When we explore the particle survival probability in certain particle diameter size range $[d_1, d_2]$, we can define a dimensionless survival parameter $P$ affecting the particle survival probability (Kulmala et al., 2017):

$$P = \frac{\text{CS}'}{\text{GR}'}. \qquad (7)$$

In this work, we used the equations described above to calculate theoretical formation rates, particle survival probabilities and condensation sinks at the measurement sites. The average CS values for each NPF event were calculated for the same time





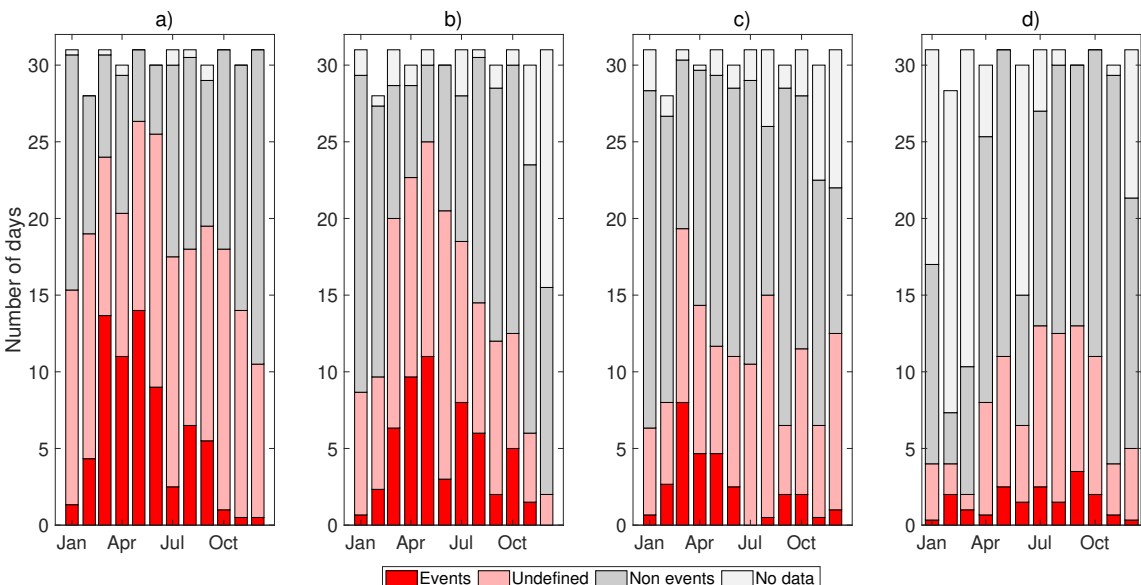

**Figure 2.** The average number of event days, non event days, undefined days and days with missing data during every month in a) Hyytiälä (January 2017- June 2019), b) Värriö (January 2017- June 2019), c) Tomsk (January 2017- June 2019), and d) ZOTTO (September 2006-April 2009).

periods as the average particle formation rates. To quantify the differences in growth rates, condensation sinks and particle survival probabilities between the measurement sites, we determined the survival parameter $P$.

## 3   Results and discussion

### 3.1   Frequency of new particle formation events

The average number of NPF event days, non-event days, undefined days, as well as days with missing or bad data for each
month in Hyytiälä, Värriö, Tomsk and ZOTTO are shown in Fig. 2. It can be seen that in Hyytiälä, there were more event days than at the other three sites. In Hyytiälä, NPF events were most frequent during spring time, which is in agreement with earlier observations (Nieminen et al., 2014). For example, in May, there were 14 NPF event days on average, whereas in December the average number of event days was 0.5. In Värriö and Tomsk, there were also more events during spring than in other seasons. In Värriö, the average number of event days per month varied from 11 in May to zero in December and in Tomsk from 8 days
in March to zero in July. In ZOTTO, the events appear to be almost evenly distributed throughout the year, excluding winter, when there were very few events. The monthly average number of event days in ZOTTO varied from 3.5 in September to 0.3 in February and December. The NPF event classification in ZOTTO was done based on particle size distributions (PNSD) measured at 50 m height. When comparing PNSD data measured at 50 m and 300 m in ZOTTO, we found that on some days NPF events could only be observed at 50 m, while on other days events could be seen at both 50 m and at 300 m. Undefined





**Table 1.** Geometric mean and median values of condensation sinks (CS), growth rates ($GR_{3-25}$, $GR_{10-25}$ and $GR_{15-40}$) and formation rates ($J_3$, $J_{10}$ and $J_{15}$) in Hyytiälä, Värriö, Tomsk and ZOTTO. Condensation sinks and formation rates are calculated for event times.

| | | CS $(10^{-3}\,\mathrm{s}^{-1})$ | $GR_{3-25}$ $(\mathrm{nm\,h}^{-1})$ | $GR_{10-25}$ $(\mathrm{nm\,h}^{-1})$ | $GR_{15-40}$ $(\mathrm{nm\,h}^{-1})$ | $J_3$ $(\mathrm{cm}^{-3}\mathrm{s}^{-1})$ | $J_{10}$ $(\mathrm{cm}^{-3}\mathrm{s}^{-1})$ | $J_{15}$ $(\mathrm{cm}^{-3}\mathrm{s}^{-1})$ |
|---|---|---|---|---|---|---|---|---|
| Hyytiälä | mean | 1.4 | 1.9 | 2.0 | 2.3 | $2.3\cdot10^{-1}$ | $1.3\cdot10^{-1}$ | $9.0\cdot10^{-2}$ |
| | median | 1.4 | 1.7 | 2.0 | 2.1 | $2.6\cdot10^{-1}$ | $1.5\cdot10^{-1}$ | $8.6\cdot10^{-2}$ |
| Värriö | mean | 0.76 | 1.9 | 2.2 | 2.3 | $7.7\cdot10^{-2}$ | $6.7\cdot10^{-2}$ | $5.4\cdot10^{-2}$ |
| | median | 0.72 | 1.9 | 2.2 | 2.4 | $7.9\cdot10^{-2}$ | $7.3\cdot10^{-2}$ | $6.6\cdot10^{-2}$ |
| Tomsk | mean | 4.1 | 1.2 | 1.3 | 1.4 | $3.6\cdot10^{-1}$ | $1.8\cdot10^{-1}$ | $7.6\cdot10^{-2}$ |
| | median | 4.3 | 1.3 | 1.3 | 1.3 | $4.1\cdot10^{-1}$ | $1.8\cdot10^{-1}$ | $8.7\cdot10^{-2}$ |
| ZOTTO | mean | 1.5 | - | - | 1.4 | - | - | $1.3\cdot10^{-1}$ |
| | median | 1.5 | - | - | 1.4 | - | - | $9.4\cdot10^{-2}$ |

days were most frequent in ZOTTO during autumn, whereas at the other sites the undefined days had a similar seasonal pattern with event days. In ZOTTO, on the contrary to other places, there were also so called "night events" which started in the late evening or close to midnight instead of the typical morning event times. The characteristics of different event types observed in ZOTTO are discussed in more detail in Sect. 3.3.

### 3.2    Event characteristics

The statistics of particle formation and growth rates at different sizes ranges and CS determined for the event times are presented in Table 1 for each measurement site. In Hyytiälä, there were 189 NPF events during January 2017−June 2019 of which particle GR and $J$ could be calculated for 87 event days (33 in 2017, 37 in 2018 and 17 in 2019). The median GR on event days was 1.7 nm h$^{-1}$ and median $J_3$ during event times was 0.26 cm$^{-3}$s$^{-1}$. The median CS during event times was $1.4\cdot10^{-3}$ s$^{-1}$.

   In Värriö, there were 149 NPF events during January 2017−June 2019 of which particle GRs and $J$s could be calculated for 190    65 event days (21 in 2017, 32 in 2018 and 12 in 2019). The median GR on event days was 1.9 nm h$^{-1}$ and median $J_3$ during event times was 0.079 cm$^{-3}$s$^{-1}$. The median CS during event times was $0.72\cdot10^{-3}$ s$^{-1}$.

   For Tomsk, particle growth rates and formation rates could be calculated for 79 event days (21 in 2017, 29 in 2018 and 29 in 2019). The median GR was 1.25 nm h$^{-1}$, and the median $J_3$ was 0.41 cm$^{-3}$s$^{-1}$. The median CS during event times was $4.3\cdot10^{-3}$ s$^{-1}$.

The modefits, GRs, coagulation sinks and $J$s were also calculated for Hyytiälä, Värriö and Tomsk using particle size distributions only starting from 10 nm or from 15 nm. Thus, we got also the values for $GR_{10-25nm}$, $GR_{15-40nm}$, $J_{10}$ and $J_{15}$.

   In ZOTTO, the formation and growth rates were calculated for all event days (44 in total). The median GR was 1.35 nm h$^{-1}$. The median formation rate $J_{15}$ during event times was 0.094 cm$^{-3}$s$^{-1}$. The median CS during the events was $1.5\cdot10^{-3}$ s$^{-1}$.





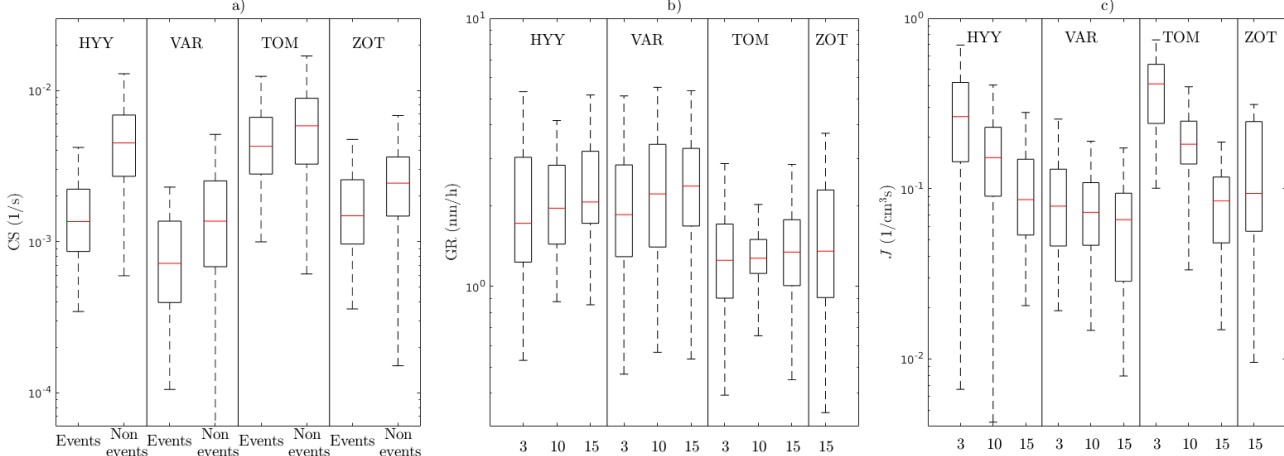

**Figure 3.** a) The CSs ($s^{-1}$) in Hyytiälä (Jan 2017-Jun 2019), Värriö (Jan 2017-Jun 2019), Tomsk (Jan 2017-Jun 2019), and ZOTTO (Sep 2006 - Apr 2009) during non event days (averages of each non event day) and event days (averages during event times), b) GRs (nm h$^{-1}$) of event days in Hyytiälä (Jan 2017- Jun 2019), Värriö (Jan 2017-Jun 2019), Tomsk (Jan 2017-Jun 2019), and ZOTTO (Sep 2006 - Apr 2009) and c) formation rates ($J_3$ averages during event times, cm$^{-3}$s$^{-1}$) in Hyytiälä (Jan 2017- Jun 2019), Tomsk (Jan 2017-Jun 2019), Värriö (Jan 2017-Jun 2019) and ZOTTO (Sep 2006 - Apr 2009). The boxes describe the 25 % and 75 % percentiles and the red horizontal line describes the median of the data set. The whiskers indicate other data range, excluding outliers.

Figure 3 illustrates CSs during non-event days (means and medians of the whole day) and event times, and GRs and $J$s for different size ranges in Hyytiälä, Värriö, Tomsk and ZOTTO. CS is the highest in Tomsk and the lowest in Värriö. The low CS in Värriö is reasonable, since in the regions surrounding Värriö there is very little population, and thus anthropogenic emissions of particles and trace gases are low. At all measurement sites, CS is higher during non-event days than during event times, which is consistent with previous studies (Kyrö et al., 2014; Nieminen et al., 2014).

The growth rates are the highest in Hyytiälä and Värriö and lower in Tomsk and ZOTTO. On average, growth rate increases with increasing particle size, which is consistent with previous observations (Yli-Juuti et al., 2011). Most clearly this size-dependency is seen in Hyytiälä and Värriö, whereas in Tomsk the differences between growth rates at different sizes are less pronounced.

The formation rates are the highest in Tomsk and Hyytiälä and the lowest in Värriö and ZOTTO. The formation rates are smaller at larger sizes, which is due to the losses of particles while they grow, as illustrated also by Eq. 5. The values of $J_{15}$ at different measurement sites are quite similar: their median values are within a factor 1.4 from each other. We can also see, that in the formation rate values in ZOTTO, there is more variation than in other places.

According to these results, CS and particle formation rate cannot fully explain infrequent NPF events in ZOTTO. Particle growth rates in ZOTTO are low, but on the other hand, they do not differ much from GR in Tomsk.





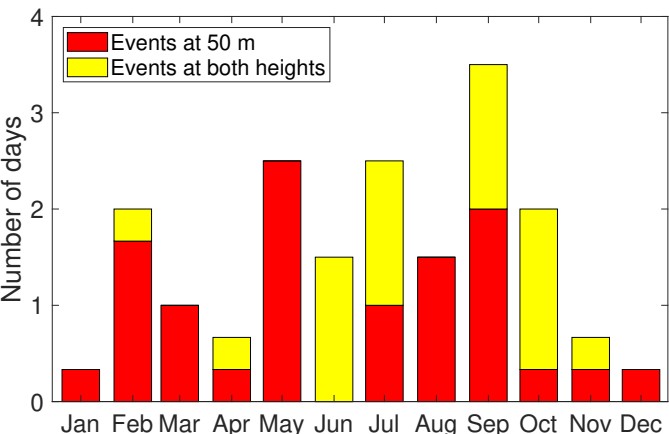

**Figure 4.** The average number of NPF event days during every month in ZOTTO. Events observed only at 50 m are shown with red and events observed at both heights (50 m and 300 m) are shown with yellow.

## 3.3 Event types in ZOTTO

The event classification was made for the whole PNSD data set of ZOTTO dividing days to event days, non event days, undefined days, and days with no available data. Because some of the NPF events in ZOTTO could only be seen in data measured at 50 m height, the event days were further divided to "only 50 m events" and "events seen at both heights" to analyze the possible differences between them. The seasonal distribution of these event types is shown in Fig. 4. There were 17 days when the event could be seen at both heights, and 27 days when it could be seen only at 50 m. Events seen at both heights were

most frequent during autumn time. The boxplot of event start times and event growth rates and formation rates for both event types in ZOTTO are shown in Fig. 5. There clearly are differences related to whether event has been observed in both heights or only at 50 m. The events observed only at 50 m are more typically night-time events (start time is later in the day around 17:00 and the growth of the particles can be followed into the next morning), and on average their growth and formation rates are smaller than in events observed at both heights (geometric mean of GR is 1.7 nm h$^{-1}$ and $J$ is 0.11 cm$^{-3}$ s$^{-1}$ for events

observed at both heights, while the corresponding values for events observed only at 50 m are 1.3 nm h$^{-1}$ and 0.09 cm$^{-3}$ s$^{-1}$). However, it should be noted that in the formation rates this difference is not significant. In fact, the highest formation rates at ZOTTO are observed in the events occurring only at 50 m height. The lower GR at 50 m could be related to event start time; lower GR events are typically also night events when the concentrations of condensing vapours could be lower than during daytime. The events occurring at both heights are also likely representing NPF at a larger area, since the footprint of the air

sampled at 300 m is much larger than at closer to the ground. The further analysis about the reasons causing the differences is outside of the scope of this paper.





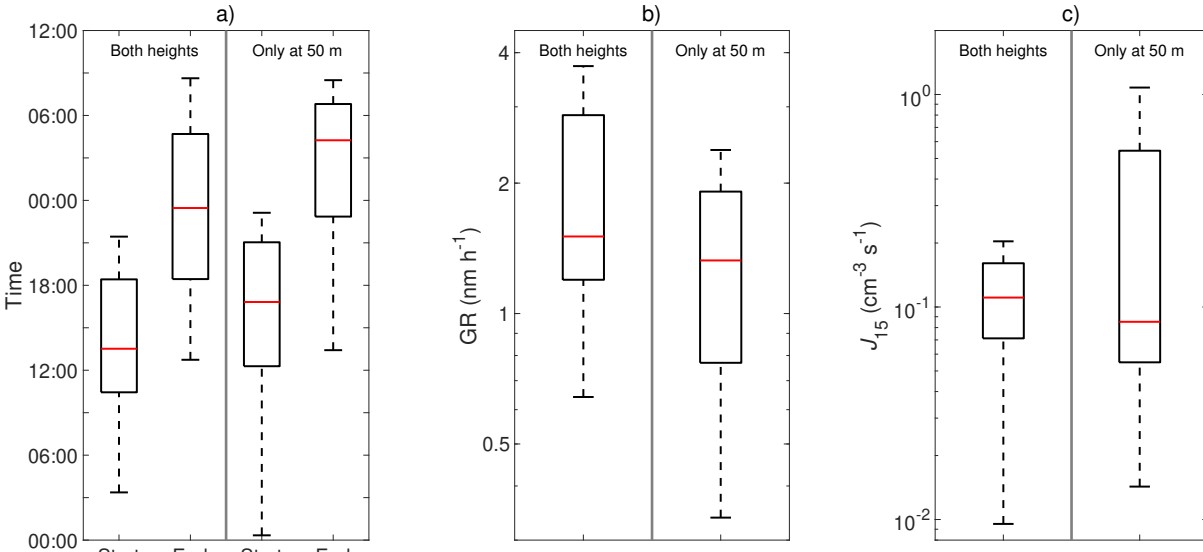

**Figure 5.** Characteristics of events observed only at 50 m and at both heights 50 m and 300 m in ZOTTO: a) start and end times of the events, b) particle growth rates and c) particle formation rates. The boxes describe the 25 % and 75 % percentiles and the red horizontal line describes the median of the data set.

### 3.4 Effect of minimum observed particle size

In Fig. 6, there are examples of NPF events observed in Hyytiälä, Värriö and Tomsk. In the left-hand side panels (a, c, e), the particle size-distribution data is shown from the smallest measured particle diameter (3 nm) and in the right-hand side panels
(b, d, f), the same PNSD data is shown starting from the larger particle size (15 nm). It can be seen that in all cases, it is more difficult to observe the NPF event using only PNSD data starting at 15 nm. In Hyytiälä, the example day in Fig. 6a has two events, one starting around noon and another one later in the afternoon. The earlier event can be identified also in Fig. 6b based on particle data starting from 15 nm, but very few particles from the later event reach above 15 nm. Missing the PNSD data in the size range 3-15 nm affects also the modefits (the geometric means of the fitted modes are shown in Fig. 6 with black dots),
which are used for calculating particle GRs.

These examples clearly show how the minimum observed particle size can affect the observed NPF event frequency. When the minimum observed particle size is larger, many of the NPF events can be undetected. Some of the event days are possibly classified to undefined days (due to uncertainty in observing the first steps of particle formation and growth) and some undefined days could be classified to non-event days (if there are no signs of new particles appearing above the minimum observed particle
size). Thus, when investigating atmospheric NPF, it is essential to measure particle number size-distributions starting from as small particle size as possible (preferably from 3 nm or below).

In Fig. 7, there are examples of one event day in ZOTTO observed at the heights of 50 m and 300 m. On this day, the event can be seen at both heights, but it is clearer at the height of 50 m due to higher particle concentrations which makes the event



**Figure 6.** In a) NPF event day (1.3.2018) in Hyytiälä is presented starting from particle diameter 3 nm. In b), the particle sizes 3-15 nm are removed from the data and the event is presented upwards from the particle diameter 15 nm. In c), NPF event day (23.4.2019) in Värriö station is presented upwards from particle diameter 3 nm and d) 15 nm. In e) NPF event day (9.3.2018) in Tomsk Fonovaya station is presented upwards from particle diameter 3 nm and in f) the event is presented upwards from the particle diameter 15 nm. White horizontal line represents the particle diameter 25 nm, and the black markers describe the modefit points.





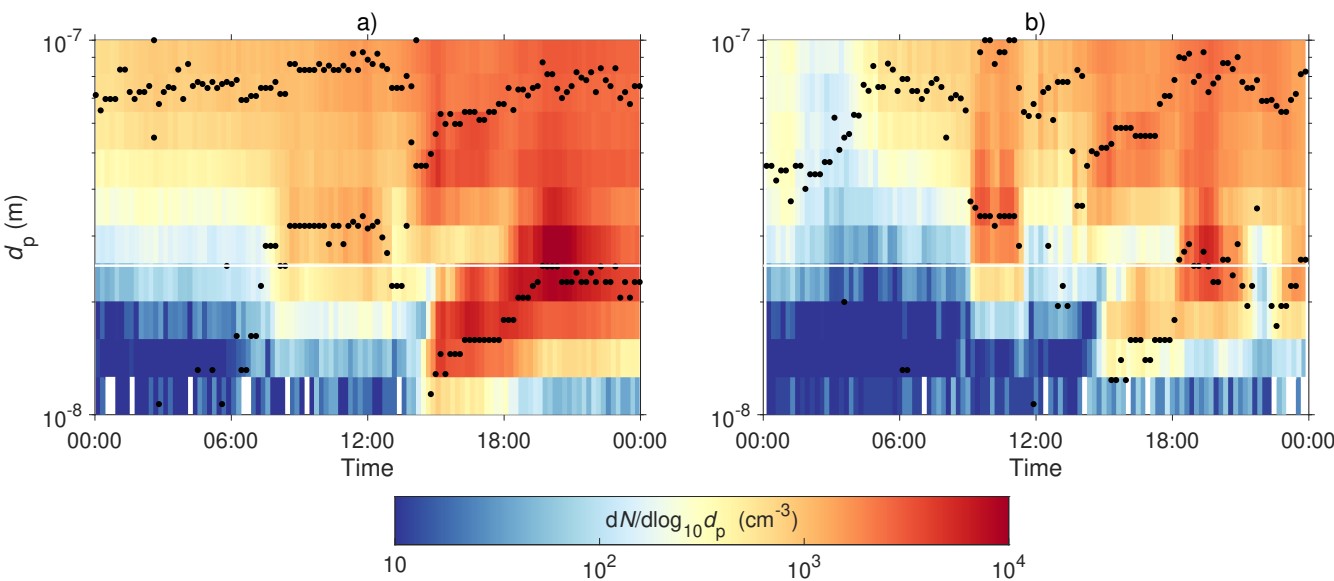

**Figure 7.** The NPF event on 27.2.2007 in ZOTTO observed at the height of a) 50 m, and b) 300 m above ground.

observable at smaller sizes than at 300 m. Also the growth of the particles is more continuous at 50 m (based on the mode fits)
than at 300 m.

## 3.5 Effect of particle survival probability

Figure 8a presents theoretical particle survival probabilities $J_3/J_{1.5}$, $J_{10}/J_{1.5}$ and $J_{15}/J_{1.5}$ as a function of the survival parameter $P$ = CS'/GR' and Fig. 8b shows $J_3/J_{1.5}$ and $J_{15}/J_{1.5}$ as a function of GR and CS. The probability of a freshly-formed particle to survive from 1.5 nm to 10 and 15 nm is considerably smaller than from 1.5 nm to 3 nm. When survival
parameter $P$=30 (a typical survival parameter value in Tomsk), $J_3/J_{1.5}$ is 10 % but $J_{10}/J_{1.5}$ is only 2 % and $J_{15}/J_{1.5}$ only 1.6 %. Figure 8b shows the theoretical survival probabilities and CS and GR values observed during NPF events at the different measurement sites. It also illustrates which magnitude GR and CS cause negligibly low survival probabilities. For example, if GR = 0.1 nm h$^{-1}$ and CS = 0.001 s$^{-1}$, the survival probability is only 0.001 for 3 nm particles. The survival probabilities are the lowest in the upper left corner of Fig. 8b where growth rate is low and condensation sink is high. When comparing the
measured survival probabilities at the four sites, it can be seen that the survival probabilities are lowered in Tomsk due to high CS and low GR and in ZOTTO due to low GR. However, the survival probabilities in ZOTTO and Hyytiälä are pretty similar, so the infrequent NPF occurrence in ZOTTO cannot be entirely explained by the low particle survival probability.





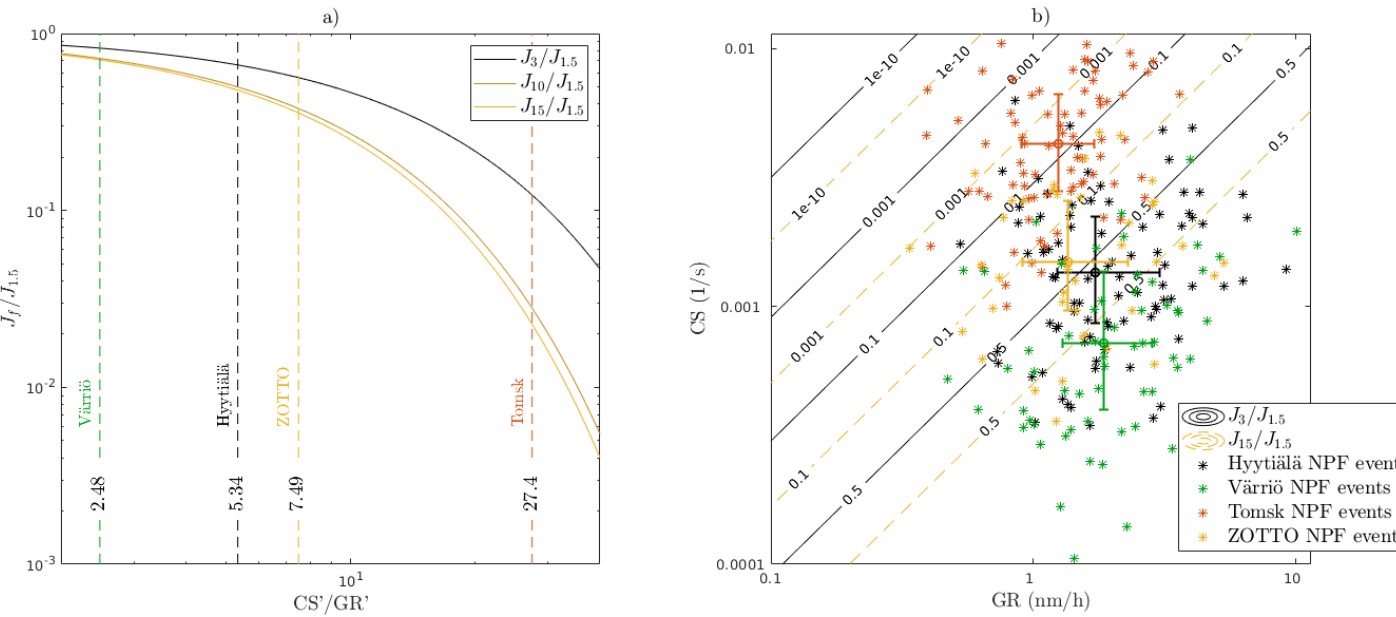

**Figure 8.** The theoretical particle survival probabilities a) $J_3/J_{1.5}$, $J_{10}/J_{1.5}$ and $J_{15}/J_{1.5}$ as a function of the survival parameter $P$=CS'/GR' and b) $J_3/J_{1.5}$ and $J_{15}/J_{1.5}$ as a function of GR and CS. The dashed lines indicate the median values of survival probabilities at each site based on the observed CS and GR. In b), the data points corresponding to CS and GR observed during NPF events in Hyytiälä, Värriö, Tomsk, and ZOTTO are shown by asterisks. The studied particle size is above 3 nm in Hyytiälä, Värriö and Tomsk and 15 nm in ZOTTO. The error bars show the 25% and 75% percentiles of the data points and the circles are the medians of the data points.

Figure 9 presents the relative number of event days compared to the sum of event and non event days observed at each site as a function of theoretical particle survival probability. For Hyytiälä, Värriö and Tomsk the survival probabilities were calculated

with the larger diameter ($d_2$) of 3 nm, 10 nm and 15 nm (the used smaller diameter ($d_1$) is 1.5 nm). Undefined days and days with bad or missing data were excluded from this comparison for the better comparability between the sites. In order to obtain the relative number of event days also from observations of above 10 nm and 15 nm particles, the event classification was made again with only PNSD data starting from 10 nm and 15 nm, respectively. This allows us to see if there are NPF events that cannot be observed based on only PNSD starting from these larger particle sizes. In this figure, we have included only the event

days when the parameters could be determined for all sizes (3 nm, 10 nm and 15 nm) for the better comparability of survival probabilities. The growth rates $GR_{3-25nm}$ have been used for calculating all survival probabilities of Hyytiälä, Värriö, and Tomsk, while for ZOTTO GRs are $GR_{15-40nm}$.

From Fig. 9, we can see that the larger the minimum detectable particle size, the lower the particle survival probability to this size is. This is reasonable, since the survival probability of freshly-formed particles to a certain size decreases, when this

size increases (Figure 8a). Still, in Hyytiälä and Värriö the difference in survival probabilities at 10 nm and 15 nm is rather small compared to Tomsk. In addition, the number of observed event days is smaller when the minimum particle size is larger.



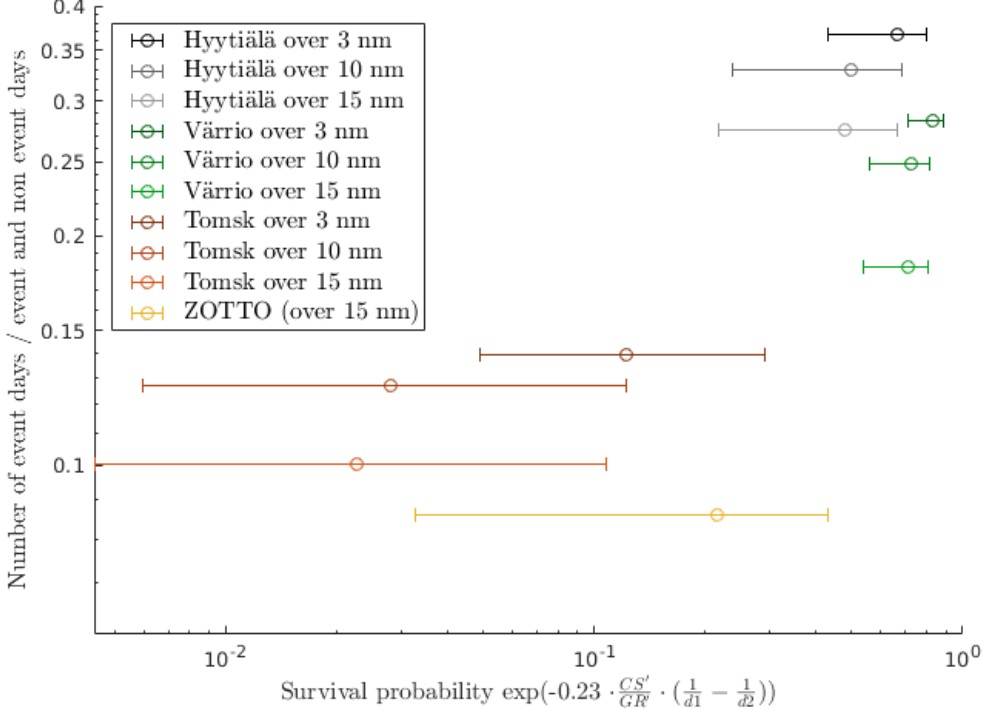

**Figure 9.** The relative number of event days compared to event and non event days as a function of a particle survival probability. The points show the median of event day survival probabilities in Hyytiälä (Jan 2017 – Jun 2019), Tomsk (Jan 2017– Jun 2019) and Värriö (Jan 2017 – Jun 2019). The horizontal bars describe 25 % and 75 % percentiles of the survival probabilities. Survival probabilities for ZOTTO are calculated using Tomsk GR and CS values and the particle size $d_2$=15 nm. In all of the survival probability calculations the smaller particle diameter $d_1$ = 1.5 nm.

Generally, the fraction of NPF event days increases with the increasing particle survival probability. When looking at the whole data set, the observed number of NPF events also increases with particle survival probability. This illustrates, how the number of observed NPF events depends on the smallest observed particle size, when part of the events are left unnoticed if the data of

smaller particle size is missing.

## 4   Conclusions

The main objective of this work was to find the reasons why there are only few NPF events observed in ZOTTO compared to other boreal forest stations in Hyytiälä, Värriö, and Tomsk. To investigate this, we studied NPF characteristics by calculating particle survival probabilities and GR, CS and particle formation rates for NPF events observed at these four measurement

stations. Generally, there are two possible reasons why only few events can be observed in ZOTTO. First, NPF events may





really be infrequent in ZOTTO compared to other stations, because of different conditions there (such as possibly lower concentrations of precursor vapors). The second possible reason is that because of the differences between the stations in particle size distribution measurements and the smallest measured particle size, NPF events in ZOTTO appear to be infrequent, due to some of the events occurring at smaller sizes than measured at the station.

Our results illustrate that in Hyytiälä and Värriö, NPF events are more frequent than at the Siberian sites, but there is also a higher variation in the number of NPF events between different seasons. In spring time, about every other day is an event day and during other seasons, events are clearly rarer. In Tomsk, there are less NPF events and they are most frequent during spring time similar to Hyytiälä and Värriö. In ZOTTO, hardly any seasonal variation in the frequency of NPF events can be observed. On average, 37 % of days (excluding undefined and bad data days) in Hyytiälä were event days during the studied time period.

In Värriö, the fraction of event days was 28 %, in Tomsk 14 % and in ZOTTO 9 %. Here, the fraction in ZOTTO is higher (by a factor of 3) than what was observed by Wiedensohler et al. (2019), due to new event classification including night events, and excluding undefined and days with bad data.

We also investigated CS, particle growth and formation rates at these sites. CS during event times was about three times higher in Tomsk than in Hyytiälä and ZOTTO and about five times higher than in Värriö. GRs were higher in Hyytiälä and

Värriö than in Tomsk and ZOTTO. The formation rates $J_{15}$ in ZOTTO were smaller than $J_3$ in Hyytiälä and Tomsk, but if we compare ZOTTO formation rates to $J_{15}$ values in other places, the formation rates were about the same in all places (within a factor of 1.4). Thus, in ZOTTO, CS and GR were low, in Tomsk GR was low but CS high, and in Hyytiälä and Värriö CS was low but GR higher. These values affect the survival parameter $P=CS'/GR'$, which was the highest in Tomsk (median($P$) = 27), lower in ZOTTO (median($P$) = 7.5) and in Hyytiälä (median($P$) = 5.3) and the lowest in Värriö (median($P$) = 2.5). They also

affect the particle survival probability, which has a median value of 66 % in Hyytiälä, 83 % in Värriö, 22 % in ZOTTO but only 14 % in Tomsk due to low GR and high CS.

The survival probability can explain a part of the differences in the event frequencies. However, there are still fewer events in ZOTTO than in Tomsk despite the higher survival probability, so it is probable that there are other differences. In addition, in Värriö and ZOTTO a lower number of NPF events is observed at similar survival probability values than in Hyytiälä and

Tomsk. Thus, there can be some other properties affecting event frequency or particle survival probability. It should be noted that Värriö and ZOTTO are the stations where there are no major cities or population centers nearby.

Our results show that in addition to CS and GR, the minimum measured particle size can affect both the observed NPF event frequency and the calculated particle survival probability. In Hyytiälä, when the minimum particle size was increased from 3 nm to 15 nm, the observed event frequency dropped from 37 % to 28 %. In Värriö, the frequency dropped from 28 % to 18 %

and in Tomsk from 14 % to 10 % (Fig. 9). When comparing the event frequencies observed with the minimum particle size of 15 nm at Tomsk (10 %) to the event frequency at ZOTTO (9 %), there is no relevant difference between these sites anymore. Thus, the difference in the minimum observable particle sizes can explain almost all differences in event frequencies between these measurement sites. According to these results, in order to get reliable NPF analysis results, it is necessary to measure PNSD data upwards from a particle size as small as possible.





*Data availability.* Data from SMEAR stations (Hyytiälä and Värriö) is available at https://smear.avaa.csc.fi/ and the data from Tomsk-Fonovaya and ZOTTO stations is available from the authors upon request.

*Author contributions.* HU, JK, MK and TN analyzed the data, interpreted the results and wrote the manuscript. All authors provided comments and suggestions on the manuscript.

*Competing interests.* The authors declare that they have no conflict of interests.

*Acknowledgements.* We acknowledge the following projects for supporting this work: ACCC Flagship funded by the Academy of Finland (grant number 337549); Academy professorship funded by the Academy of Finland (grant number 302958); Academy of Finland projects number 1325656, 311932, 316114 and 325647; Russian Mega Grant project "Megapolis - heat and pollution island: interdisciplinary hydroclimatic, geochemical and ecological analysis" (application reference 2020-220-08-5835); INAR project "Quantifying carbon sink, CarbonSink+ and their interaction with air quality" funded by Jane and Aatos Erkko Foundation; European Research Council (ERC) project
ATM-GTP (contract number 742206). Data at Tomsk-Fonovaya was obtained using the infrastructure of the Institute of Atmospheric Optics, Siberian Branch, Russian Academy of Sciences, under state order no. AAAA-A17-117021310142-5, including the Center for Collective Use Atmosfera. The analysis of aerosol dynamics was supported by the Russian Foundation for Basic Research (grant number 19-05-50024). The technical and scientific staff in Hyytiälä, Värriö, Tomsk-Fonovaya and ZOTTO stations are acknowledged for maintaining the long-term measurements.



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
