# Peer review of "Occurrence of new particle formation events in Siberian and Finnish boreal forest"

_Atmospheric Chemistry and Physics, 2021_

## Referee Comment (RC2)

**Review comments to manuscript:  acp-2021-530**

**Occurrence of new particle formation events in Siberian and Finnish boreal forest**

**by Uusitalo H. et al.**

**General comments:**
The manuscript analyzes observations of particle number size distribution (PNSD) from four sites in boreal forest environments to compare properties of the observed new particle formation (NPF) events, such as occurrence and growth rates. Major discrepancies are found at the ZOTTO site in Siberia which shows much lower frequency of NPF events. The authors attribute these features to the minimum detectable particle size resulting from the different measurement instrument adopted.

The topic is relevant to the ACP readership, the manuscript is well written, but major concerns about the analyses presented, novelty of the work and generalizability of the findings are present, as highlighted in the specific and technical comments below, which need to be addressed to consider it suitable for publication.

**Specific comments:**

- Objectives: the objectives of the study are not clearly stated. In the introduction the authors state that they seek to understand  why "NPF is infrequently observed in Zotino" (line 52), but they fail to properly address it, by limiting the analyses to a display of the observations. However, at line 230 the authors say the "the reasons causing the differences is outside the scope of this paper". Finally, at line 282 it's stated: "The main objective of this work was to find the reasons why…", so the objectives should be clearly stated and appropriate analyses performed and presented to address them. In addition to differences in the instrumentation deployed, both meteorological and chemical conditions should be considered to explore the fundamental sources of discrepancies highlighted (particularly as the issue with the minimum detectable size is not fully provable at ZOTTO and it seems to only partially explain the observed discrepancies).
- Instrumentation differences: at line 53 and through the manuscript the authors suggest that discrepancies in the occurrence of the frequency of NPF events may derive from the too large minimum detectable size of the instrument deployed at ZOTTO. More details about the instrumentation used at the four sites should be provided in addition to the size range covered. For example, details are needed on the number of size bins, temporal resolution, accuracy and uncertainty in the measurements. As observations from different instrumentations are compared, comments about to what extent these measurements are comparable should be included.
- Observations time period: details on the four measurement sites should be provided. For example, it is not clear at which heights the analyzed PNSD are measured and if this height is the same at all sites. If there are differences, they should be accounted for (or at least discussed) when comparing the observations at the four sites. Further, it is not clear why observations from 2017-2019 are compared at three sites, while for the ZOTTO site, which is the one showing major discrepancies, data from 2006 to 2009 are analyzed. The different

time period analyzed may be a major cause of discrepancies and does not seem to be justified as observations at Hyytiala and Varrio have been collected since 1996 and 1997, respectively. A fair intercomparison of the sites would require at least to analyze the same time period. Since the observations at ZOTTO were collected about 10 years before the data at the other sites, possible changes in the emissions, transport of nucleation precursors may have occurred and should be addressed. All results displayed would be impacted and would need to be revised.

- Novelty of the work: The authors should highlight the scientific contribution and novelty of this work. As the methodology is taken from prior studies and the data are not new, more effort should be put into interpreting and analyzing the observations from the four sites in a more consistent and thorough way.

**Technical comments:**

- Line 30: this is an important point as discrepancies in the properties of NPF events could be related to differences in both meteorological conditions and chemical precursors. The literature review should be expanded to include studies beyond boreal forest environments to demonstrate how these variables play a role in dictating NPF occurrence, and how their role varies in different regions of the world. This addition would provide the background for the additional analyses needed to better interpret discrepancies in the data.
- Line 99: a classification of NPF events based on a visual inspection of the data is not ideal, as it may be subjective and not reproducible. A quantitative and automatic method to analyze PNSD should be applied.
- Line 101: it is not clear how a day is classified as an event day. For how long does a burst of new particles should last and/or for how long does it have to grow and up to which size? These details are needed for reproducibility of the results, particularly as the current event classification is qualitative.
- Line 113: what is the timestep for each set of observations? If they are different how are the observations homogenized for a proper comparison?
- Line 131: how is the coagulation coefficient K determined and how is $d'_p$ defined?
- Line 154: the conversion between $nm\ h^{-1}$ and $m\ s^{-1}$ is obvious and not needed.
- Line 181: what are the causes of this nighttime events? Could they be related to some boundary layer dynamics, or specific chemical or transport mechanisms? How do these events look like? Some of the event characteristics may point out to important mechanisms responsible for occurrence of NPF events at ZOTTO and help with the interpretability of the discrepancies with the other sites. These mechanisms should be discussed/interpreted in light of prior literature studies (e.g. Crippa, P., Petäjä, T., Korhonen, H., El Afandi, G. S., and Pryor, S. C.: Evidence of an elevated source of nucleation based on model simulations and data from the NIFTy experiment, Atmos. Chem. Phys., 12, 8021–8036, https://doi.org/10.5194/acp-12-8021-2012, 2012.)
- Line 230: basic analyses that would provide more insights and should be added include for example backtrajectory analysis to describe the "climatology" of the events in term of transport of precursors and meteorological conditions, or for specific events to show discrepancies among the sites. Could the NPF events be related to an elevated source or

transport of precursors? Sources of discrepancies could be in the instrumentation adopted, but also in the emission context (as the ZOTTO observations are collected 10 years prior the ones at the other sites), availability of precursors (are there any gas phase observations available at some of the sites or from nearby stations?).

- Figure 6 could be improved as it is confusing to have a different y-axis in the two columns, so the panels are not immediately comparable.
- Figure 7: the temporal resolution of the data at ZOTTO is clearly different from the other sites. How is this going to impact NPF detection? As mentioned earlier, for how long a burst of new particles has to grow (and to what size) to be classified as a NPF event? This may also impact the classification at ZOTTO given the different temporal and size resolution.
- Line 260-262: this sentence indicates that no conclusive findings can be inferred from the presented analyses, which therefore need to be expanded to investigate at least key chemical/dynamic processes behind NPF events.
- Figure 9: this figure has a very low quality and should be remade.
- Line 285: here the authors speculate about possible causes, which however should be the main focus of the paper and supported by analyses. A map of the sites may be included when the data are described, along with some meteorological/chemistry/emission summary to help understand the regional background of the events.
- Line 311: the authors seem to suggest the different emission context could be a reason of the discrepancies observed, which would be quite expected. This point should be supported by more data/analyses, as mentioned in previous comments.
- Line 318-319: this sentence seems to suggest that the data in ZOTTO are unreliable. More information about the data accuracy should be presented. Have other studies presented/investigated NPF mechanisms at ZOTTO? If so they should be mentioned in the introduction to provide enough background knowledge to the reader. If not, a more thorough analysis of the dataset should be included.